# Translation and Adaptation of the French Version of the Risk Stratification Index, a Tool for Stratified Care in Chronic Low Back Pain: A Pilot Study

**DOI:** 10.3390/medicina58040469

**Published:** 2022-03-23

**Authors:** Alexandra Naïr, Chiao-I Lin, Pia-Maria Wippert

**Affiliations:** 1Medical Sociology and Psychobiology, Department of Physical Activity and Health, University of Potsdam, Am Neuen Palais 10, 14469 Potsdam, Germany; anair@uni-potsdam.de (A.N.); chialin@uni-potsdam.de (C.-I.L.); 2Faculty of Health Sciences Brandenburg (Joint Faculty of the University of Potsdam, The Brandenburg Medical School Theodor Fontane and the Brandenburg University of Technology Cottbus–Senftenberg), University of Potsdam, Am Neuen Palais 10, 14469 Potsdam, Germany

**Keywords:** RSI, translation-reliability, back-pain screening, yellow flags, psychosocial moderators

## Abstract

*Background and Objectives*: Low back pain is a worldwide health problem. An early diagnosis is required to develop personalized treatment strategies. The Risk Stratification Index (RSI) was developed to serve the purpose. The aim of this pilot study is to cross-culturally translate the RSI to a French version (RSI-F) and evaluate the test-retest reliability of RSI-F using a French active population. *Materials and Methods*: The RSI was translated from German to French (RSI-F) based on the guidelines of cross-cultural adaptation of self-report measures. A total of 42 French recreational athletes (age 18–63 years) with non-specific low back pain were recruited and filled in the RSI-F twice. The test-retest reliability was examined using intraclass correlation coefficient (ICC_1,2_) and Pearson correlation coefficient. *Results*: Finally, 33 questionnaires were analyzed (14 males and 19 females, age 31 ± 10 years, 9.5 ± 3.2 h/week of training). The test-retest of RSI-F CPI and DISS were excellent (CPI: ICC_1,2_ = 0.989, *p* < 0.001; *r* = 0.989, *p* < 0.001; DISS: ICC_1,2_ = 0.991, *p* < 0.001; *r* = 0.991, *p* < 0.001), as well as Korff pain intensity (ICC_1,2_ = 0.995, *p* < 0.001; *r* = 0.995, *p* < 0.001) and disability (ICC_1,2_ = 0.998, *p* < 0.001; *r* = 0.998, *p* < 0.001). *Conclusion*: The RSI-F is linguistically accurate and reliable for use by a French-speaking active population with non-specific low back pain. The RSI-F is considered a tool to examine the evolution of psychosocial factors and therefore the risk of chronicity and the prognostic of pain. Further evaluations, such as internal, external validity, and responsiveness should be evaluated in a larger population.

## 1. Introduction

Low back pain (LBP) is a common health problem all around the world and 50–80% of adults experience LBP at some point in their lifetime [1]. LBP is defined as pain, muscle tension, or stiffness that is localized below the costal margin (inferior rib cage), above the inferior gluteal folds, and might occur with or without leg pain (sciatica) [2]. In Western and Central Europe, LBP is the leading cause of years lived with disability (years of life lived in unhealthy conditions) and may lead to long-term disability associated with recurrent use of health services, activity limitation, or work absence [3,4,5,6,7]. The prevalence of LBP is 84% and the prevalence of chronic low back pain (CLBP) is approximately 23%. In France, the prevalence of LBP was 15% whereas in Germany it was 17% [8,9]. LBP appears not only in the general population but also among athletes, notably in Martial Arts such as Judo (between 34.5% and 40.9% depending on the weight categories) or Kyokushin Karate (55%) [10,11]. 

LBP is a global health issue and the most expensive musculoskeletal problem in modern societies. However, specific treatments for LPB are missing, causing frequent recurrent LBP or transition to CLBP for many patients [5,6,7,8]. For that reason, detecting patients’ risk of LBP chronicity at an initial stage is essential. Various back pain screening tools (i.e., the STarT Back screening tool) were developed to specify if a unimodal or multimodal program is needed [12].

Moreover, patient-based biopsychosocial indicators (also known as yellow flags factors) are the best way to evaluate patients with LBP and at least five areas should be assessed: social disability, pain symptoms, work disability, well-being, and LBP-specific function [13,14,15,16,17]. Therefore, the Risk Stratification Index (RSI) was developed, a bio-psychosocial screening tool for risk estimation of low back pain chronicity, extended absenteeism, and disability after a low back pain episode [18].

This brief questionnaire covers a multiple spectrum of biopsychosocial risk factors: yellow flags (e.g., fear-avoidance, depression), orange flags (e.g., lifestyle factors, life events, and chronic stress), blue flags (e.g., social status, position, and economy), and black flags (e.g., unemployment rate, workplace environment). It also includes the measurement for pain persistence and pain intensity. The RSI was developed in German and validated using the general population, athletes, and patients from rehabilitation centers. The RSI was also translated and validated in English [19,20,21].

The advantage of the RSI compared to other back pain screeners is that the RSI assesses whether individuals at higher risk can respond to an exercise therapy or whether moderating factors would interfere with this [22]. On this basis, assignment to a unimodal or multimodal therapy is made. Furthermore, the RSI is used in combination with a prevention screener (RPI-S) to give information about an optimized individualized therapy management for each psychosocial risk area [23]. Both screeners allow an individualized therapy management in functional and psychosocial areas [18]. Another advantage of the RSI is that it can provide a high prediction accuracy even at a lower pain level, which is why it is more sensitive than other screeners such as the STarT Back [24,25,26,27]. 

In France, the High Authority of Health (HAS) usually uses three questionnaires for detecting psychosocial risk factors: the STarT Back screening tool (SBST), which offers a stratified risk management approach to unimodal or multimodal treatment, the short version of the Örebro Questionnaire (ÖMPSQ), which is predictive of absenteeism, and the Roland-Morris Questionnaire (RMDQ) for the indication of acute LBP [28,29].

However, except for SBST, no other tool can stratify patients’ risk of back pain chronification and suggest corresponding treatment (exercise only or exercise with behavior intervention) based on psychosocial factors. The RSI provides another possibility for physicians to assess the risk for CLBP because it includes the Fear Avoidance Beliefs Questionnaire (FABQ) and Depression Scale (HADS), which assesses the level of fear, apprehensions, and avoidance related to low back pain, and the level of anxiety and depression, respectively [30,31]. Therefore, the objective of this pilot study is to cross-culturally adapt and translate the RSI into French and to examine the test-retest reliability.

## 2. Materials and Methods

The current study was a cross-sectional study of a cross-cultural adaptation validation of a self-report questionnaire and was conducted between November and December 2020. Two methodological steps were carried out: the first step was a cross-cultural translation of the RSI from German to French by four translators and the second step was the evaluation of the test-retest reliability of the RSI-F. 

The German version (original) of the RSI was adapted to French according to the guidelines for the process of cross-cultural adaptation of self-report measures [32]. Six steps were applied:

Step I: Initial translation (German to French): Two bilingual translators, a medical doctor and a non-medical background translator, translated the RSI from German to French. 

Step II: Synthesis of the translation: Translations differences from the two translators were synthesized in the consensus meeting, resulting in the first version of the translated questionnaire.

Step III: Back translation: The first version of RSI-F was translated back to German by two German native speakers without any medical background. The translators were unknown to the original German RSI. 

Step IV: Expert committee review: The expert committee included two methodologists, a health professional, a French language professional, and the translators (forward and back translators). They discussed and compared the backward translations with each other and with the original questionnaire to secure the pre-final version of RSI-F.

Step V: Test of the pre-final version: Participants filled up the pre-final version of RSI-F to check the meaning and the comprehension of the subjects of each item included in the questionnaire. Any issues that occurred in the process of filling up the questionnaire were reported and adjusted to the final version of RSI-F. 

Step VI: Submission of documentation and coordination committee for appraisal of the adaptation process: The final version of RSI-F was given to the developer of the instrument who reviewed it. 

To evaluate the psychometric properties of RSI-F, *n* = 42 participants who train regularly, from a private physiotherapy clinic and two sport clubs (Krav Maga team), were contacted. 

Candidates were eligible for inclusion if they were 18 years old or older with at least one episode of non-specific LBP (≥days) in the past 12 months and native French speakers [33]. The main exclusion criteria were pregnancy, specific low back pain (i.e., fracture, cancer, infection, or inflammatory disease), non-active participants (<3 h training/week), and an inability to fill out questionnaires independently. All participants signed up for the study after receiving both written and oral information. Each participant completed the RSI-F and was questioned about any difficulties encountered in completing the questionnaire or understanding the purpose or meaning of each question. This study was approved by the Ethics Committee of the University of Potsdam within a pain network study (47/2016). 

The Risk Stratification Index: RSI aims to evaluate the estimation of prognostic of pain (disability and intensity) and risk of CLBP development. This tool contains four psychological risk variables: pain, distress, medical care-environment, and social environment. RSI was developed for the general population and further validated for athletes. RSI includes 28 items. A total of 7 items evaluates pain intensity (CPI) and disability (DISS) related to the Chronic Pain Grade questionnaire (CPG); 14 items evaluate distress; 4 items evaluate medical care-environment; and 6 items evaluate social environment [34]. Patients are classified as “low risk” of developing CLBP if RSI = 0–29, “medium risk” if RSI = 29–49, “high risk” if RSI = 50–69, and “very high risk” if RSI = 70–100 [18,19].

For the psychometric properties’ exploration, the test-retest reliability was analyzed. All the participants filled out the questionnaire twice, with a week period in between testing. The test-retest reliability of each item was examined by the intraclass correlation coefficient (ICC_1,2_) for the scale items and by Cohen’s kappa for nominal and ordinal (weighted kappa value) items. For the total score, the test-retest reliability was evaluated by ICC_1,2_ and Pearson correlation coefficient (*r*). 

ICC_1,2_ and kappa values were interpreted as follows: poor <0.40, fair 0.40–0.59, good 0.60–0.74, and excellent 0.75–1.00 [35,36]. The Pearson correlation coefficient was indicated as: very high correlated 0.90–1.00, high correlated 0.70–0.90, moderate correlated 0.50–0.70, low correlated 0.30–0.50, and negligible correlated 0.00–0.30 [37].

All data analysis was performed using IBM SPSS 25.0 (Chicago, IL, USA). Test-retest reliability for each item was examined by intraclass correlation coefficient (ICC_1,2_) and by kappa values; the total score was evaluated by Pearson correlation coefficient (*r*) and ICC_1,2_ with 95% Confidence Interval (CI). Level of significance was set at <0.05.

## 3. Results

### 3.1. Participants

After excluding participants who did not meet the inclusion criteria, 33 questionnaires could be accepted for analysis. In total, 14 males and 19 females with a mean age of 31.8 (±10.1) from recreational athletes (3–10 h training/week) to regular athletes were recruited. Among them, 28 participants were suffering from LBP at the time of the study (see Table 1).

### 3.2. Validation of the Cross-Cultural Translation

The German version of RSI was adapted to French based on the guidelines. During phase II of cross-cultural translation, two versions of forward translation were difficult to synthesize because the German and French education systems are different. Therefore, the educational-related questions were adapted to the French education system. No other specific problem of experiential, idiomatic, semantic, or conceptual equivalence was relevant during the translation process. 

### 3.3. Psychometric Properties

Test-retest reliability: The RSI-F CPI/DISS and the French version of the CPG (CPG-F) CPI/DISS were measured at baseline and one week later showing excellent total scores. The results are shown in Table 2. For each item, test-retest reliability was good to excellent (ICC_1,2_ values ranged from 0.921 to 1.00 with *p* < 0.001 and kappa values from 0.812 to 1.00). Results for each item are shown in Table 3.

## 4. Discussion

In this pilot study, a bio-psychosocial screening tool, the Risk Stratification Index was translated and cross-culturally adapted from German to French and the test-retest reliability within an active population with low back pain was evaluated. Results showed an overall excellent test-retest reliability, meaning that the RSI-F can be used in clinical practice for persons suffering from low back pain. 

The test-retest reliability from the RSI-F (ICC_1,2_ = 0.989–0.991) is in line with the German version which showed a good overall reliability (RSI-DISS: ICC_1,2_ = 0.74; RSI-CPI: ICC_1,2_ = 0.721, unpublished data). The RSI translation performs very well in comparison with the French version of the ÖMPSQ (ICC_1,2_ = 0.89 (95% CI: 0.79–0.95)), the Finnish ÖMPSQ (ICC_1,2_ ranged from 0.59 to 0.96), the Brazilian ÖMPSQ (ICC_1,2_ = 0.76 (95% CI: 0.28–0.89), and the Persian version of ÖMPSQ (ICC_1,2_ ranged from 0.49 to 0.87; ICC_1,2_ = 0.82) [38,39,40]. Moreover, it is in line with translated versions of other back pain related questionnaires such as the French version of the SBST (ICC_1,2_ = 0.90 (95% CI: 0.81–0.95)), and the RMDQ (ICC_1,2_ = 0.89 (95% CI: 0.83–0.93)) [41,42,43]. 

Although the CPG is a part of the RSI (see Table 2 CPG-F and Table 3 items Korff 2–7), it was of further interest how the translation of the CPG performs: the presented data show a very good test-retest reliability of CPG-F which is comparable to the Spanish CPG version (ICC_1,2_ = 0.81 (95% CI: 0.684–0.891)) and the Turkish CPG version (ICC_1,2_ = 0.972 (95% CI: 0.94–0.98)) [44,45].

The current pilot study applied the common one-week interval to examine the overall test-retest reliability. Previous studies also applied the same time frame: this was the case for the translation of the French version of the Keele STarT MSK Tool (ICC_1,2_ = 0.97), the Brazilian version of the ÖMPSQ (ICC_1,2_ = 0.76 (95% CI: 0.28–0.89), and the French version of the RMDQ (ICC_1,2_ = 0.89 (95% CI: 0.83–0.93) [41,42,46].

The original version of the RSI progressed from 205 predictors and showed a precise estimation of the expected individual CPG-CPI and DISS values. The original RSI showed an excellent strength (Area Under the Curve (AUC) = 0.81, 95% CI: 0.76–0.86) to discriminate patients at a risk of developing greater score of CPI and an acceptable strength (AUC = 0.74, 95% CI: 0.63–0.85) to discriminate patients at a risk to develop a greater score of DISS. In contrast, PickUp (predicting pain intensity) evolved from 20 predictors and showed an acceptable performance (AUC = 0.66), the SBST contained 9 predictors for pain disability with an excellent performance (AUC = 0.92), and the ÖMPSQ contained 24 predictors for disability and return to work [18]. 

Screening of patients’ psychosocial risk factors and stratified allocation to treatment are needed since different types of treatment for low back pain are recommended. In addition to the traditional neuromuscular training, prevention guidelines for non-specific LBP indicate that biopsychosocial factors and psychological treatment options should be considered [47]. Multimodal treatments including psychosocial interventions are favored over unimodal approaches because they address factors including distress, social environment, or subjective pain [47]. Consequently, available tools for screening and stratifying patients’ psychosocial risk are important [48,49,50]. 

A few back specific instruments have been cross culturally adapted to French (i.e., the ÖMPSQ, the SBST, and the RMDQ) but no “gold standard” questionnaire exist to measure the biopsychosocial factors, and therefore the risk of chronicity in low back pain [35,38,39]. Thus, RSI-F showed to be a good new tool to assess the chronicity of low back pain.

The strength of the current study is the use of two different fighting clubs and one private physiotherapist clinic, allowing the recruitment of a broad spread of patients with LBP with a fair representation of both sexes. Moreover, the main scales of CPG included in the RSI were also translated and evaluated. This questionnaire is one of the most important measurements worldwide and from what is known, no French version has previously been translated and validated. The main limitation of this study was the small sample size, but the statistical precision of the study was good: the lower end of the confidence interval of the ICC is above the minimum accepted level for reliability of 0.7 [51]. Additional testing of the psychometric properties of the French version of the RSI questionnaire is needed to assess the predictive validity (cut off scores for risk groups); it still needs to be confirmed in a broader French population and sample size.

## 5. Conclusions

In conclusion, the French version of the RSI questionnaire is reliable, linguistically accurate, and acceptable for use by French-speaking patients. The next step will be to assess other psychometric properties (e.g., cutoff score and responsiveness) of RSI-F to effectively examine the evolution of psychosocial factors, the risk of chronicity, and the prognostic of pain in clinical standard routines in a larger sample.

## Figures and Tables

**Table 1 medicina-58-00469-t001:** Baseline characteristics of the study population.

Sociodemography	Baseline Characteristics
*n* (men/women)	33 (14/19)
Age (years, M ± SD)	31.8 ± 10.1
Training (hours per week, M ± SD)	9.5 ± 3.2
Regular athletes (*n* (%))	15 (45)
Recreational athletes (*n* (%))	18 (55)
LBP at the time of the study (*n* (%))	28 (85)
Risk Stratification Index CPI (*n* (%))	
Low risk	13 (40%)
Medium risk	16 (48%)
High risk	4 (12%)
Risk Stratification Index DISS (*n* (%))	
Low risk	30 (90%)
Medium risk	3 (10%)

*n*, sample size; M mean; SD, standard deviation; h, hours; LBP, low back pain; CPI, characteristic pain intensity, scale of Chronic Pain Grade questionnaire; DISS, subjective pain disability, scale of Chronic Pain Grade questionnaire.

**Table 2 medicina-58-00469-t002:** Descriptive statistics of RSI-F CPI/DISS and CPG-F CPI/DISS at baseline and retest.

Items		Baseline	Retest	ICC_1,2_	95% CI	*r*	*p*-Values
RSI-F	CPI	31.3 ± 14.2	31.6 ± 13.8	0.989	0.977–0.994	0.989	<0.001
	DISS	16.0 ± 9.8	15.9 ± 9.9	0.991	0.982–0.996	0.991	<0.001
CPG-F	CPI	31.0 ± 21.7	30.8 ± 21.4	0.995	0.990–0.997	0.995	<0.001
	DISS	18.7 ± 21.7	18.6 ± 21.7	0.998	0.996–0.999	0.998	<0.001

ICC: intraclass correlation coefficient; CI: confidence interval; RSI-F: French version of the Risk Stratification Index; CPG-F: French version of the Chronic Pain Grade; CPI: characteristic pain intensity, scale of Chronic Pain Grade questionnaire; DISS: subjective pain disability, scale of Chronic Pain Grade questionnaire; *r*, Pearson correlation coefficient.

**Table 3 medicina-58-00469-t003:** Test-retest reliability for each item.

Category	Items	Baseline	Retest	Type of Data	Reliability Test	ICC (95% CI); Kappa
		Mean ± SD	Mean ± SD			
Pain	Korff 2	1.79 ± 2.20	1.88 ± 2.12	scale	ICC	0.989 (0.994–0.997) ***
	Korff 3	4.33 ± 2.78	4.24 ± 2.84			0.997 (0.997–0.999) ***
	Korff 4	3.18 ± 2.13	3.12 ± 2.13			0.997 (0.993–0.998) ***
	Korff 5	1.36 ± 1.92	1.39 ± 1.92			0.998 (0.996–0.999) ***
	Korff 6	1.97 ± 2.30	1.94 ± 2.29			0.999 (0.997–0.999) ***
	Korff 7	2.27 ± 2.71	2.24 ± 2.70			0.999 (0.998–0.999) ***
Distress	RSI 8	0.36 ± 0.65	0.45 ± 0.67	scale	ICC	0.948 (0.887–0.973) ***
	RSI 9	2.76 ± 0.50	2.70 ± 0.53			0.940 (0.879–0.970) ***
	RSI 10	2.85 ± 0.71	2.85 ± 0.67			0.967 (0.933–0.984) ***
	RSI 11	2.64 ± 1.54	2.64 ± 1.52			0.993 (0.987–0.997) ***
	RSI 12	3.82 ± 1.81	3.76 ± 1.82			0.995 (0.991–0.998) ***
	RSI 13	2.12 ± 2.19	2.15 ± 2.21			0.995 (0.990–0.998) ***
	RSI 14	1.85 ± 1.15	1.82 ± 1.16			0.983 (0.965–0.991) ***
	RSI 14a	1.73 ± 1.21	1.73 ± 1.21			0.968 (0.934–0.984) ***
	RSI 15	1.45 ± 0.97	1.42 ± 0.97			0.975 (0.950–0.988) ***
	RSI 16	1.70 ± 0.92	1.67 ± 0.99			0.974 (0.948–0.987) ***
	RSI 17	1.24 ± 0.97	1.21 ± 0.96			0.992 (0.984–0.996) ***
	RSI 18	1.33 ± 1.08	1.27 ± 1.07			0.987 (0.973–0.993) ***
	RSI 18a	1.18 ± 0.73	1.12 ± 0.70			0.969 (0.938–0.985) ***
	RSI 19	1.24 ± 0.79	1.27 ± 0.76			0.921 (0.921–0.981) ***
Medical care-environment	RSI 6	1.61 ± 0.50	1.61 ± 0.50	nominal	Kappa	1.00 ***
	RSI 6a	0.30 ± 0.47	0.30 ± 0.47			1.00 ***
	RSI 6b	0.09 ± 0.29	0.09 ± 0.29			1.00 ***
	RSI 6c	0.33 ± 0.48	0.33 ± 0.48			1.00 ***
Social environment	RSI 3	7.52 ± 1.12	7.52 ± 1.12	ordinal	Weighted kappa	1.00 ***
	RSI 4	4.94 ± 3.12	4.94 ± 3.12	nominal	Kappa	1.00 ***
	RSI 5	7.55 ± 1.56	7.45 ± 1.58	scale	ICC	0.991 (0.981–0.995) ***
	RSI 7	1.21 ± 0.99	1.15 ± 1.00	nominal	Kappa	0.812 (0.721–0.949) ***

SD: Standard deviation; ICC: intraclass correlation coefficient; CI: confidence interval; RSI-F: French Risk Stratification Index; ***: *p* < 0.001.

## Data Availability

Not applicable.

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
