# Peer review of "Translation and Adaptation of the French Version of the Risk Stratification Index, a Tool for Stratified Care in Chronic Low Back Pain: A Pilot Study"

_medicina, 2022, doi:10.3390/medicina58040469_

Round 1
Reviewer 1 Report
An interesting work. However, I have a few comments: ABSTRACT: Well written Introduction: Well written MATERIAL AND METHODS: Please explain why validation was only performed on healthy people. Why was there no validation in patients with back pain? RESULTS: Well written DISCUSSION: Please explain why validation was only performed on healthy people. Why was there no validation in patients with back pain?Author Response
Please see the attachment.

Reviewer 2 Report
Was the selection of the survey sample representative? Personally, I feel that it is very simple, and the sample size is too small to be used for statistical analysis. Please explain. I suggest to expand the survey sample object and quantity, at least 10 times more than the questionnaire items.
Thanks.
Reviewer 3 Report
Review Letters
Dear Authors
The title of this study seems to be consistent with the Medicina. I think this paper will be better if some minor and major points are corrected.
Minor points
Line 22: There must be a space between numbers and symbols.
Line 23 to 25: All "p" presented in this paper are statistical symbols and should be converted to italics.
Line 39: When entering the number of references, three or more consecutive references should be marked with '-'. For example, [3,4,5,6] → [3-6].
Line 46: When entering the number of references, three or more consecutive references should be marked with '-'. For example, [5,6,7] → [5-7].
Line 54: When entering the number of references, three or more consecutive references should be marked with '-'. For example, [11,12,13,14] → [11-14].
Line 65: When entering the number of references, three or more consecutive references should be marked with '-'. For example, [16,17,18] → [16-18].
Line 74: When entering the number of references, three or more consecutive references should be marked with '-'. For example, [22,23,24,25] → [22-25].
Line 79: When entering the number of references, three or more consecutive references should be marked with '-'. For example, [26,27,28] → [26-28].
Line 165: The unit for age given in Table 1 is missing.
Line 199: When entering the number of references, three or more consecutive references should be marked with '-'. For example, [37,38,39] → [37-39].
Line 229: When entering the number of references, three or more consecutive references should be marked with '-'. For example, [47,48,49] → [47-49].
Major points
Line 87-88: The authors presented the purpose of the study in Line 87-88 as "...the objectives of this study were to translate the RSI into French, confirming its translation validity and to examine the test-retest reliability." However, when looking at the results and discussion of this paper, the contents and results of 'validity' are omitted. Is there any special reason for this?
Line 173 to 176: The authors indicated the same values in Table 2 in the text. This is redundant. In the following sentence, "Test-retest reliability: Test-retest of total score was excellent for RSI-F CPI (ICC1,2 = 0.989 (0.977-0.994), p<0.001; r = 0.989, p<0.001), RSI -F DISS (ICC1,2 = 0.991(0.982-0.996), p<0.001; r = 0.991), p<0.001), CPG-F CPI (ICC1,2 = 0.995(0.990-0.997), p<0.001; r = 0.995, p<0.001), CPG-F DISS (ICC1,2=0.998(0.996-0.999), p<0.001; r=0.998, p<0.001)" is preferably deleted.
The discussion is a bit awkward, and it would be good to re-state the whole by inserting the results of validity.
Sincerely,
Round 2
Reviewer 2 Report
I have no problem. thank you.
Author Response
Dear Reviewer,
thank you once again for your previous feedbacks.
Kind regards,
Alexandra Naïr
Reviewer 3 Report
Re-review Letters
Dear Authors
It is believed that the authors had a lot of trouble revising the paper. The revised article is much better than the previous one. However, there are some minor errors and minor corrections are required.
Minor points
Line 24-26: The "**" displayed between lines 24 to 26 should be deleted.
Line 193: "**" must be deleted.
Line 194: "table" must be modified to "Table".
Line 214: Since Table 2 gives a p-value of <0.001, the "**" is unnecessary and should be deleted. It is also recommended to delete "** p < 0.001: significant" in the footnote of the table.
Table 3: In general, the * sign is as follows. * means p < 0.05, ** means p < 0.01, and *** means p < 0.001. In Table 3, ** should be replaced with ***.
In references, abbreviations of sources must be marked with “.”.
Author Response
Dear Reviewer,
thank you again for your suggestions and your time.
Minor points
Line 24-26: The "**" displayed between lines 24 to 26 should be deleted.
Response 1: This is revised based on the comment. Please see Line 24-26
Line 193: "**" must be deleted.
Response 2: This is revised based on the comment. Please see Line 204
Line 194: "table" must be modified to "Table".
Response 3: This is revised based on the comment. Please see Line 205
Line 214: Since Table 2 gives a p-value of <0.001, the "**" is unnecessary and should be deleted. It is also recommended to delete "** p < 0.001: significant" in the footnote of the table.
Response 4: This is revised based on the comment. Please see Table 2 and the footnote of Table 2.
Table 3: In general, the * sign is as follows. * means p < 0.05, ** means p < 0.01, and *** means p < 0.001. In Table 3, ** should be replaced with ***.
Response 5: This is revised based on the comment. Please see Table 3.
In references, abbreviations of sources must be marked with “.”.
Response 6: This is revised based on the comment.
Thank you once again